# NET Proteome in Established Type 1 Diabetes Is Enriched in Metabolic Proteins

**DOI:** 10.3390/cells12091319

**Published:** 2023-05-05

**Authors:** Samal Bissenova, Darcy Ellis, Aïsha Callebaut, Guy Eelen, Rita Derua, Mijke Buitinga, Chantal Mathieu, Conny Gysemans, Lut Overbergh

**Affiliations:** 1Clinical and Experimental Endocrinology (CEE), Department of Chronic Diseases and Metabolism (CHROMETA), KU Leuven, 3000 Leuven, Belgium; 2Laboratory of Angiogenesis and Vascular Metabolism, Department of Oncology, KU Leuven, 3000 Leuven, Belgium; 3Laboratory of Angiogenesis and Vascular Metabolism, Center for Cancer Biology, VIB, 3000 Leuven, Belgium; 4Laboratory of Protein Phosphorylation & Proteomics, Department Cellular & Molecular Medicine, KU Leuven, 3000 Leuven, Belgium; 5SyBioMa, Proteomics Core Facility, KU Leuven, 3000 Leuven, Belgium; 6Department of Nutrition and Movement Sciences, Maastricht University, 6211 LK Maastricht, The Netherlands; 7Department of Radiology and Nuclear Medicine, Maastricht University Medical Center, 6229 HX Maastricht, The Netherlands

**Keywords:** neutrophils, NETosis, NET proteome, type 1 diabetes, metabolism

## Abstract

Background and aims: Type 1 diabetes (T1D) is a chronic autoimmune disease characterized by a T-cell-mediated destruction of the pancreatic insulin-producing beta cells. A growing body of evidence suggests that abnormalities in neutrophils and neutrophil extracellular trap (NET) formation (NETosis) are associated with T1D pathophysiology. However, little information is available on whether these changes are primary neutrophil defects or related to the environmental signals encountered during active disease. Methods: In the present work, the NET proteome (NETome) of phorbol 12-myristate 13-acetate (PMA)- and ionomycin-stimulated neutrophils from people with established T1D compared to healthy controls (HC) was studied by proteomic analysis. Results: Levels of NETosis, in addition to plasma levels of pro-inflammatory cytokines and NET markers, were comparable between T1D and HC subjects. However, the T1D NETome was distinct from that of HC in response to both stimuli. Quantitative analysis revealed that the T1D NETome was enriched in proteins belonging to metabolic pathways (i.e., phosphoglycerate kinase, glyceraldehyde-3-phosphate dehydrogenase, and UTP-glucose-1-phosphate uridylyltransferase). Complementary metabolic profiling revealed that the rate of extracellular acidification, an approximate measure for glycolysis, and mitochondrial respiration were similar between T1D and HC neutrophils in response to both stimuli. Conclusion: The NETome of people with established T1D was enriched in metabolic proteins without an apparent alteration in the bio-energetic profile or dysregulated NETosis. This may reflect an adaptation mechanism employed by activated T1D neutrophils to avoid impaired glycolysis and consequently excessive or suboptimal NETosis, pivotal in innate immune defence and the resolution of inflammation.

## 1. Introduction

Innate immune cells, such as macrophages and dendritic cells, are implicated in the initiation and perpetuation of type 1 diabetes (T1D), the most common metabolic disease in children and adolescents [1]. Recent evidence also adds neutrophils and neutrophil-derived substances as players in T1D pathophysiology [2,3]. Neutrophils were detected within the pancreas of both pre-symptomatic autoantibody-positive and newly diagnosed T1D subjects with active disease [2]. Yet, neutrophil function in subjects with established T1D could also be dysregulated, leading to a weakened ability to overcome infections and persistent inflammation [4,5]. Whether this is due to an intrinsic neutrophil defect or to the diabetic microenvironment remains to be elucidated.

While neutrophils were believed to be short-lived terminally differentiated innate phagocytes, recent evidence suggests that they display great plasticity depending on the inflammatory environment [6]. They employ various effector functions such as degranulation, phagocytosis and oxidative burst, in addition to the formation of neutrophil extracellular traps (NETs), termed NETosis, that are mesh-like structures consisting of decondensed chromatin entangled with anti-microbial granule proteins and proteases, such as myeloperoxidase (MPO), neutrophil elastase (ELANE), and proteinase 3 (PR3) [7]. These physiological functions may be aberrant or suboptimal in inflammatory and autoimmune disorders, in which inappropriate recruitment of neutrophils may lead to tissue damage and chronic inflammation. Excessive NET formation has been associated with active disease in anti-neutrophil cytoplasmic antibody (ANCA)-associated vasculitis (AAV), systemic lupus erythematosus (SLE), and rheumatoid arthritis (RA) [8,9]. However, contradicting findings on the levels of peripheral blood neutrophils, NETosis, and NET markers have been described in T1D and warrant further investigation [5,10,11]. Moreover, the kinetics and mechanisms of NETosis differ depending on the infectious or inflammatory stimuli [12]. Phorbol 12-myristate 13-acetate (PMA), lipopolysaccharide (LPS), and various types of bacteria induce lytic NETosis within 2–4 h, which depends on a nicotinamide adenine dinucleotide phosphate (NADPH)-mediated oxidative burst and MPO-dependent migration of granular ELANE to the nucleus where it cleaves histone proteins [13]. On the other hand, calcium ionophores, such as ionomycin, lead to what is termed ‘vital’ NETosis within minutes and appear to act independent of NADPH oxidase but rely primarily on calcium mobilization and peptidyl arginine deiminase 4 (PADI4) activation, leading to citrullination of histones and chromatin decondensation [14]. Various studies demonstrated that NETs induced by different stimuli were heterogeneous in terms of protein composition, suggesting that NETs in different pathological conditions may have specific biological implications [15]. NETs isolated from individuals with SLE and RA had distinct proteome profiles (e.g., MPO, leukocyte elastase inhibitor, and thymidine phosphorylase (TYMP) in SLE, while RNASE2 in RA) [15]. Moreover, the NETome of SLE patients with lupus nephritis (LN) was enriched in annexin A1 and α-enolase compared to those without LN [16].

Here, we studied the protein composition of NETing neutrophils (NETome) from people with established T1D and sex- and age-matched healthy controls (HC) under both basal and PMA- or ionomycin-stimulated conditions. Current data suggest that neutrophils in established T1D individuals have a distinct NETome enriched in glycolytic proteins when activated with both NADPH oxidase-dependent and -independent stimuli, which may allow better usage of glucose for neutrophil survival and function.

## 2. Research Design and Methods

### 2.1. Human Subjects and Ethics Statement

Peripheral blood was collected between 8:00 and 11:00 a.m. through venipuncture from T1D and HC donors recruited at the University hospital UZ Leuven. Subjects signed informed consent; all experiments involving human subjects were approved by the Ethics Committee Research UZ/KU Leuven (S62381). Demographics for T1D and HC donors in the study are shown in Table 1. All T1D donors were on intensive insulin therapy consisting of either multiple daily injections with insulin analogues or insulin pump usage as determined by a clinician at the University hospital UZ Leuven. To avoid interference by acute inflammation and glycemic dysregulation at the time of diagnosis on neutrophil behavior, we selected people with established T1D (between 4 and 38 years post-diagnosis).

### 2.2. Isolation of Primary Human Neutrophils

Peripheral blood samples from T1D and sex- and age-matched HC donors were collected in EDTA-coated tubes. Granulocyte counts were determined using ABX automated hematology analyzer (Horiba, Kyoto, Japan). Neutrophils were isolated by centrifuging whole blood over a LymphoPrep™ gradient (Stem Cell, Cat No. 07851, Saint Égrève, France) for 30 min at 600× *g*, resulting in the separation of the different cell types according to cell density. The sediment containing the neutrophils and erythrocytes was then resuspended in 1% gelatin solution (Sigma, Cat No. G-9382, St. Louis, MO, USA) and incubated for 30 min. The top layer containing the neutrophils was collected, and hypotonic lysis was performed using filtered MilliQ water to eliminate residual erythrocytes. All of the described procedures were conducted at room temperature and under sterile conditions.

### 2.3. Quantification and Visualization of NETs by Immunofluorescence

Isolated neutrophils were seeded on 0.01% poly-l-lysine (Sigma, P4832) coated coverslips at 500,000 cells/mL. The cells were then stimulated or not with 100 nM of PMA (Sigma, Cat No. P8139) or 20 µM of ionomycin (Sigma, Cat No. I0634) for 3 h at 37 °C in 5% CO_2_. The coverslips were washed, and the cells were fixed with 4% paraformaldehyde (Klinipath/VWR international, Cat No. 4177; Amsterdam, The Netherlands) for 15 min. Cells were then stained with 8 µM of Hoechst 33342 (Molecular Probes, Cat No. 33258, Eugene, OR) and mounted on slides. The percentage of NET-releasing cells (at least 500 cells per slide, quantified using a Nikon Eclipse TI microscope using the 40× objective) was calculated by normalizing the total amount of cells. Images were recorded on a Zeiss LSM 780–SP Mai Tai HP DS confocal DS Cell and Tissue Imaging Cluster (CIC) microscope using the 63× objective (supported by Hercules AKUL/11/37 and FWO G.0929.15 to Pieter Vanden Berghe, KU Leuven, Leuven, Belgium). Image processing, including the conversion of imaged z-stacks into maximum intensity projections, was carried out using Huygens (Scientific Volume Imaging, Hilversum, The Netherlands) and ImageJ/Fiji software (Java; NIH, Bethesda, MD, USA).

### 2.4. Plasma NET Markers Measurements

Plasma from T1D and HC donors was stored at −80 °C within 30 min of blood sampling. The levels of MPO and ELANE were determined using the R-plex human myeloperoxidase assay from Meso Scale Discovery (MSD, Cat No. K1514ER-2, Gaithersburg, MD, USA) and the human neutrophil elastase ELISA kit (Abcam, Cat No. ab119553), respectively, following the manufacturer’s instructions.

### 2.5. Multiplex Cytokine Assays

To assess cytokine levels in the plasma of T1D and HC donors, the human V-PLEX Proinflammatory Panel 1 kit (MSD, Cat No. K15052D-1), which measures interferon gamma (IFNγ), interleukin 1 beta (IL-1β), interleukin 6 (IL-6), and tumor necrosis factor alpha (TNF-α), was used. MSD plates were analyzed on the MESO QuickPlex SQ120 (MSD), and data analysis was performed on the MSD Discovery Workbench 3.0 (MSD).

### 2.6. NET Protein Harvesting

We optimized NET induction and harvesting (summarized in Appendix A) in order to minimize downstream contamination and obtain an optimal amount of protein for LC-MS/MS analysis by using a low amount of serum and developing an optimal NET protein harvesting technique. Neutrophils were seeded at 8 × 10^6^ cells/well in NET culture medium (RPMI 1640 medium without phenol red (ThermoFisher, Waltham, MA, USA, Cat No. 11835-063) supplemented with 25 mM HEPES and 2% heat-inactivated AB human serum (Biowest, Cat No. S00HE1040C, Nuaille, France)) in 6-well cell culture plates and stimulated with 100 nM PMA or 20 µM ionomycin at 37 °C in 5% CO_2_ for 3 h. After incubation, the supernatant was collected and stored at −80 °C. The wells were washed with serum-free NET culture medium, after which NETs were digested with 10 U/mL DNase I (Roche, Cat No. 4716728001, Basel, Switzerland) in NET culture medium containing the incubation buffer (Roche, Cat No. 4716728001) and EDTA-free protease inhibitor (Roche, Cat No. 05892791). Digestion was stopped with 5 mM EDTA pH 8.0 (Invitrogen, Cat No. AM92606, Waltham, MA, USA) and the supernatant containing NET proteins was harvested. Centrifugation was performed at 400× *g* for 10 min at 4 °C to eliminate residual cells and again at 16,000× *g* for 10 min at 4 °C to eliminate debris. The supernatants containing NET proteins were transferred into tubes containing 3 times the volume of ice-cold 100% acetone and incubated overnight at −20 °C. Precipitated and purified proteins were spun down at 10,000× *g* for 30 min at 4 °C, the acetone was gently removed and protein pellets were briefly air-dried and stored at −80 °C. Prior to downstream applications, pellets were dissolved in a 7 M Urea buffer. The protein concentrations were determined using the Pierce BCA Protein analysis kit (ThermoFisher, Cat No. 23227).

### 2.7. LC-MS/MS Analysis

NET proteins were reduced, alkylated, precipitated, and digested as described [17]. The resulting peptides were prepared in loading buffer (0.1% formic acid (FA) and 5% ACN), separated on an Ultimate 3000 Ultra Performance Liquid Chromatography system (UPLC, Dionex, Thermo Scientific) and analyzed by a hybrid quadrupole-Orbitrap Q Exactive mass spectrometer operated in a data-dependent acquisition mode (Thermo Scientific). Peptides were identified using Mascot (Matrix Science, Singapore) using Uniprot *homo sapiens* (194,620 entries) as a database. Oxidation (M) was included as a variable modification, whereas carbamidomethylation (C) was included as a fixed modification. Two missed cleavages were allowed for the trypsin digestion, and peptide tolerance was set at 10 ppm and 20 mmu for MS and MS/MS, respectively. Progenesis software (Nonlinear Dynamics Ltd., Newcastle, UK) was used for relative quantification of proteins using normalized abundance values, and Scaffold 4 (Proteome Software Inc., Portland, OR, USA) was used to validate MS/MS-based peptide and protein identifications. Peptide and protein identifications were accepted to achieve an FDR of less than 1.0%. From the resulting protein list, only those proteins that were present in at least 3 out of 4 or 2 out of 3 replicates from either T1D or HC, respectively, were considered to be quantifiable proteins.

### 2.8. Bioenergetic Profile of Neutrophils

Peripheral neutrophils isolated from donors were seeded at 2.5 × 10^6^ in 300 µL/Seahorse XFe24 sensor cartridge well (coated with Cell-Tak Cell (Corning, Cat No. 734-1081, Corning, NY, USA)) in the corresponding supplemented XF medium and incubated for 20 min at 37 °C without CO_2_ (workflow in Appendix A). For the oxygen consumption rate (OCR), XF RPMI assay medium (Agilent, Cat No. 103576-100) was supplemented with 1 mM L-glutamine, 5 mM glucose (Merck, Cat No. 108997), and 1 mM sodium pyruvate (Gibco, Cat No. 11360-039), whereas for the Seahorse XF Glycolysis stress test, the medium was supplemented with 1 mM L-glutamine. The pH of both media was adjusted to 7.4. OCR and the extracellular acidification rate (ECAR) was measured at 6 min intervals. After the first five measurements, a medium containing the vehicle, PMA (100 nM) or ionomycin (20 µM), was injected. Additionally, for ECAR measurements, glucose (5 mM) and oligomycin (1 µM) were sequentially injected. OCR and ECAR data were normalized to total protein content determined using the Pierce BCA Protein analysis kit. ‘Glycolysis’ was calculated as the difference between the average of normalized glucose-induced ECAR levels and the average of normalized stimuli- (PMA, ionomycin) or control medium-induced ECAR levels. ‘Glycolytic capacity’ was calculated as the difference between the average of normalized oligomycin-induced ECAR levels and the average of normalized stimuli- (PMA, ionomycin) or control medium-induced ECAR levels.

### 2.9. Lactate Measurements

The levels of lactate (mg/dL) were measured in the cell supernatants of unstimulated, PMA- and ionomycin-stimulated neutrophils of T1D and HC donors using the Beckman Coulter DxC700 AU automated chemistry system (Beckman Coulter, Brea, CA, USA).

### 2.10. Bioinformatic and Statistical Analyses

Principal component analysis (PCA) was performed using Prism software 9 (GraphPad, La Jolla, CA, USA). Components were selected using parallel analysis. Venn diagrams were made using the InteractiVenn platform [18]. Protein–protein interaction networks were made with Cytoscape 3.9 with manual annotation using UniProt and PANTHER 16.0 classification system (gene ontology; GO) [19]. Heat maps were made with Qlucore Omics Explorer 3.6. Statistical analysis was performed on Prism software 9 (GraphPad) using the unpaired two-tailed Mann–Whitney test or unpaired two-tailed *t*-test or Kruskal–Wallis test with Dunn’s post hoc test. A *p*-value of <0.05 was considered statistically significant.

## 3. Results

### 3.1. PMA and Ionomycin Induce Comparable Levels of NETosis in Neutrophils Isolated from T1D and HC Donors

Granulocyte counts in the peripheral blood of established T1D donors were not significantly different from those of sex-, age-, and race-matched HC subjects (Appendix A). Moreover, plasma values of inflammatory cytokines implicated in the pathophysiology of T1D, such as IFNγ, IL-1β, IL-6, and TNF-α, were comparable between T1D and HC donors (Figure 1A), as were the plasma levels of the NET constituents ELANE and MPO (Figure 1B,C).

We next investigated the levels of NET formation under both basal and PMA- or ionomycin-stimulated conditions. At baseline (unstimulated), the proportion of neutrophils undergoing spontaneous NETosis was not significantly different between T1D and HC donors, reaching 7 (±1) versus 3 (±3)%, respectively, after 3 h of incubation (*p* = 0.057). Stimulation with PMA (100 nM) increased the rate of NETosis to comparable levels in neutrophils of T1D and HC donors up to 90 (±7) and 85 (±6)%, respectively (*p* = 0.628). The natural calcium ionophore ionomycin (20 µM) induced NETosis in 77 (±16) and 64 (±2)% of neutrophils of T1D and HC donors, respectively (*p* = 0.400) (Figure 1D). Staining with the DNA-intercalating dye Hoechst identified mesh-like DNA structures, indicative of the formation of NETs, in PMA- or ionomycin-stimulated HC neutrophils, compared to mostly intact polylobal nuclei in unstimulated neutrophils (Figure 1E). While both stimulated conditions presented with a characteristic loss of the polylobal nuclear morphology, PMA-stimulated NETs had a morphology congruent with cell swelling, whereas ionomycin induced long filaments of DNA protrusions. Collectively, these results show that PMA and ionomycin induced morphologically distinct NETs at comparable frequencies in neutrophils of T1D and HC donors.

### 3.2. NET Proteomes of Peripheral Neutrophils Are Different upon PMA and Ionomycin Stimulation

Comparing the two stimuli, initially without considering disease status, we saw that in combined HC and T1D samples, PMA and ionomycin induced 370 and 321 NET proteins, respectively, with 214 NET proteins overlapping, 156 exclusive to PMA-stimulated neutrophils and 107 proteins exclusive to ionomycin-stimulated neutrophils (Figure 2A). The vast majority of PMA- and ionomycin-induced NET constituents were shared and included granular proteins such as MPO, ELANE, and proteins involved in chromatin decondensation such as PADI4 (Table 2). Both lists also contained proteins commonly released from neutrophils during NET formation, such as the inflammatory S100 proteins, neutrophil defensin, and cathepsin G, confirming the successful isolation of the NET fraction.

According to gene ontology (GO) analysis, the cellular compartment from which these NET proteins originated was comparable in response to either PMA or ionomycin stimulation, with 15–16% of NET proteins localized in the membrane, 52–57% in the cytoplasm/cytoskeletal compartment, 12–14% in the organelle compartment, 12–20% in the extracellular compartment, and 1% in the nucleus (Figure 2B). Moreover, GO analysis of the molecular functions of the identified NET proteins demonstrated that both stimuli induce proteins involved in cytoskeletal reorganization and protein binding functions (Figure 2C,D), which is consistent with cytoskeletal reorganization required for NETosis. However, proteins exclusive to the NET proteome of PMA-stimulated neutrophils were primarily involved in molecular functions associated with purine biosynthesis, such as guanyl nucleotide binding, GTPase activity, and GTP binding (Figure 2C), while proteins exclusive to the NET proteome of ionomycin-stimulated neutrophils were primarily associated with endoplasmic reticulum (ER) stress response such as unfolded protein binding and protein folding chaperones (Figure 2D). While the majority of the NET proteins induced by PMA and ionomycin were shared, those exclusive to the NETome induced by each stimulus point to their different underlying mechanisms of action.

### 3.3. PMA- and Ionomycin-Induced NET Proteomes of T1D Neutrophils Are Distinct from Those of HC

We then focused on the differences in NET proteome between T1D and HC neutrophils in response to PMA or ionomycin. Following stimulation with PMA, a total of 370 proteins were identified in the NETing neutrophils of T1D and HC subjects (full list in Appendix A; <1.0% FDR). Principal component analysis (PCA) of these proteins showed a distinct clustering of biological replicates for T1D and HC subjects (Figure 3A). Among the list of identified proteins, 278 overlapped between T1D and HC subjects, whereas 22 and 70 proteins were exclusive to T1D and HC subjects, respectively (Figure 3B). GO analysis of the proteins exclusive to either T1D or HC NETomes revealed that these proteins originated from the extracellular and organelle compartments and were involved in common molecular functions such as protein binding and catalytic activity. However, proteins exclusive to the T1D NETome originated from the cytoplasmic compartment, whereas those exclusive to the HC NETome were found to originate from the membrane. Proteins exclusive to the T1D NETome were involved in transporter activity, whereas those exclusive to the HC NETome were involved in ATP-binding and cytoskeletal motor activities (Figure 3B). The proteins exclusive to the NET proteome of either T1D or HC subjects were mapped onto protein–protein interaction networks, where they were clustered according to primary biological processes (GO annotation; Figure 3C,D). Despite being involved in similar processes, such as metabolism, gene expression, cellular homeostasis, immune function, and cytoskeletal organization, the proteins exclusive to either T1D or HC NETomes were different. For instance, we identified metabolic proteins such as isocitrate dehydrogenase (IDH1) and nicotinamide adenine dinucleotide kinase (NADK) in the NETome of T1D subjects (Figure 3D). In contrast, the NETome of HC subjects contained metabolic proteins, such as phosphoglucomutase-2 (PGM2) and aldehyde dehydrogenase family 16 member A1 (ALDH16A1), along with various proteins involved in lipid metabolism (e.g., apolipoprotein B, APOB; polyunsaturated fatty acid 5-lipoxygenase, ALOX5; apolipoprotein B receptor, APOBR) (Figure 3C).

In the ionomycin-stimulated conditions, 321 proteins were identified in NETing neutrophils of T1D and HC subjects (full list in Appendix A; <1.0% FDR). PCA of these proteins showed a distinct clustering of biological replicates for T1D and HC subjects (Figure 4A). Among the list of identified proteins, 247 were overlapping, and 21 and 53 were exclusive to the NET proteomes of T1D and HC subjects, respectively (Figure 4B). According to GO analysis, proteins exclusive to the T1D NETome were involved in molecular functions such as structural molecule activity and catalytic activity, whereas proteins exclusive to HC NETome were implicated in molecular functions such as translation and transcription regulation. Similar to the PMA-stimulated conditions, these proteins originated primarily from the extracellular and organelle compartments, while certain proteins exclusive to the NETomes of either T1D or HC originated from the membrane and cytoplasmic compartments, respectively (Figure 3B and Figure 4B). GO annotations of protein–protein interaction networks revealed different proteins belonging to similar biological functions, such as gene expression, cellular homeostasis, immune system, and cytoskeletal organization (Figure 4C,D). In the NETome exclusive to HC neutrophils, we identified proteins involved in glucose metabolism, such as 4-alpha-glucanotransferase (AGL) and ATP-dependent 6-phosphofructokinase (PFKL), as well as those involved in oxidative phosphorylation (OXPHOS; e.g., ALDH16A1, APOB). Despite their involvement in the same molecular functions and biological processes, the proteins of the T1D NETome were different from those of the HC NETome.

### 3.4. Enzymes Involved in the Glucose Metabolism Pathway Are More Abundant in the NET Proteomes of T1D Subjects Compared to Those of HC Subjects

To further investigate the particularities of the NET proteomes of T1D and HC subjects, we quantified the proteins of NETing neutrophils of both subject groups. Following stimulation with PMA, a total of 44 proteins were differentially expressed (*p* < 0.05, <1.0% FDR) between neutrophils of T1D and HC subjects (Figure 5A, full list in Appendix A). Hierarchical clustering of normalized abundance values of differentially expressed NET proteins showed that 35 proteins were enriched in NETing neutrophils of T1D subjects and nine in those of HC subjects (Figure 5B). Among those decreased in the T1D NETome were proteins involved in innate immunity, such as protein S100-A6 (S100A6, 0.44-fold), ELANE (0.30-fold), azurocidin (AZU1, 0.33-fold) and protein S100-P (S100P, 0.53-fold), with significantly lower abundance values compared to those of HC subjects (Figure 5A,C). Metabolic proteins, such as glyceraldehyde-3-phosphate dehydrogenase (GAPDH; 2.07-fold), phosphoglycerate kinase (PGK1; 1.70-fold), fructose-bisphosphate aldolase A (ALDOA; 1.96-fold), and pyruvate kinase PKM (PKM, 1.77-fold), had significantly higher abundance values in the T1D NETome compared to that of HC subjects (Figure 5A,D).

Upon ionomycin stimulation, a total of 27 proteins were differentially expressed (*p* < 0.05, <1.0% FDR) in NETing neutrophils of T1D and HC subjects (Figure 6A, full list in Appendix A). Among these, twenty and seven proteins were enriched in NETing neutrophils of T1D and HC subjects, respectively, as shown by hierarchical clustering of their normalized abundance values (Figure 6B). Similarly to the PMA-stimulated condition, proteins involved in innate immunity such as alpha-1-antichymotrypsin (SERPINA3, 0.24-fold), protein AMBP (AMBP, 0.64-fold), peptidoglycan recognition protein 1 (PGLYRP1, 0.67-fold) and proteasome subunit beta (PSMB2, 0.74-fold) were all significantly less abundant in the T1D NETome compared to those of HC subjects (Figure 6A,C). Those enriched in the T1D NETome were metabolic proteins, such as GAPDH (1.98-fold), PGK1 (1.83-fold), UTP-glucose-1-phosphate uridylyltransferase (UGP2; 1.70-fold), and phosphoglycerate mutase 1 (PGAM1, 1.74-fold) (Figure 6A,D).

Using Reactome analysis, we annotated the proteins enriched in either the T1D or HC NETomes in response to PMA or ionomycin according to the pathways in which they are involved (Figure 7). Amongst others, the NET proteins more abundant in HC compared to T1D neutrophils were associated with innate immunity in either PMA or ionomycin-stimulated conditions (Figure 7A,B). Proteins enriched in the T1D NETome in response to either PMA or ionomycin were shown to be primarily involved in glycolysis, gluconeogenesis, and glucose metabolism pathways, amongst others (Figure 7C,D). These results showed that the NETome of T1D neutrophils was significantly enriched in proteins involved in glucose metabolism, with lower abundances in proteins implicated in innate immunity.

### 3.5. PMA- and Ionomycin-Stimulated Neutrophils of T1D Subjects Have Similar Metabolic Profiles as HC

As pathway analysis pointed towards possible metabolic alterations in T1D NETing neutrophils, we studied their metabolic profile compared to that of HC. In response to PMA, there was a comparable increase in mitochondrial respiration, reported as the OCR, in neutrophils of T1D and HC subjects, with minimal responsiveness to ionomycin stimulation (Figure 8A). On the other hand, glycolysis, reported as the ECAR, was increased in both PMA- and ionomycin-stimulated neutrophils of T1D and HC subjects. This increase was again comparable between T1D and HC neutrophils (Figure 8B). Moreover, both T1D and HC neutrophils had a similar increase in glycolytic capacity in response to PMA, with no significant increase in response to ionomycin (Figure 8C). Lactate levels in the supernatant of T1D and HC neutrophils were analogous at baseline (unstimulated), as well as in response to PMA or ionomycin (Figure 8D), further corroborating the ECAR data. Overall, the bioenergetic profiling of T1D and HC neutrophils revealed no significant differences in mitochondrial respiration or glycolysis between the two subject groups.

## 4. Discussion

Neutrophils and NETosis have been implicated in different aspects of T1D pathophysiology [2,20,21,22,23]. The current study demonstrates that levels of circulating neutrophils, in addition to the frequencies of basal and stimulated peripheral NETing neutrophils and plasma NET products in T1D and HC individuals, are not significantly different. However, conflicting data exist regarding these neutrophil measurements in individuals genetically at risk for or diagnosed with T1D. While some studies reported peripheral neutrophilia and increased levels of NET markers (i.e., MPO, ELANE, PR3, PAD4, cell-free DNA-histone complexes, etc.), others showed an increase in peripheral neutrophil counts and decreased levels of NET markers [3,5,10,24,25]. The heterogeneity of the data could be related to the various disease stages. Moreover, disease presentation and duration may have an impact on neutrophil functions. Whether these observations are related to a primary neutrophil defect that would be apparent during all disease stages or to environmental cues present during active disease is unknown. Here, we opted to study peripheral-blood-derived neutrophils isolated from individuals with established T1D.

To our knowledge, this is the first study exploring the full NETome of stimulated neutrophils isolated from people with established T1D by proteomic analysis. In line with the findings of Chapman et al., the NETomes of PMA-stimulated neutrophils differed from those induced by ionomycin [15]. Despite the fact that the two stimuli induced largely the same NET proteins, some key proteins exclusive to each stimulus highlighted their different underlying mechanisms of action. PMA-induced NET proteins primarily originated from the cytoplasmic and membrane compartments and were involved in purine metabolism, which is essential to neutrophil activation [26,27]. In fact, guanine nucleotides and GTPases, more abundant in PMA-induced NETs, regulate NAPDH oxidase-dependent reactive oxygen species (ROS) formation [28,29]. On the other hand, NET proteins induced by ionomycin stimulation were shown to be involved in ER stress responses and originated primarily from the cytoplasmic and extracellular compartments. This is consistent with calcium ionophores inducing calcium mobilization through ER stress, which is essential for NADPH oxidase-independent NET formation [14,30,31]. Morphological differences in NET formation in immunofluorescent images of NETs induced by the two stimuli confirm these mechanistic differences and are congruent with previous studies [12].

Recent observations indicate that NET protein composition may be disease-specific, which has been shown in RA and SLE [15,16]. This is also the case for T1D, as our current data showed that the T1D NETome differed considerably from that of HC. Proteins involved in cytoskeletal organization, gene expression, cellular homeostasis, immune system, as well as metabolic proteins were identified. In relation to the latter, neutrophils are primarily glycolytic and require glycolytic ATP for NET formation [32,33]. Interestingly, proteins involved in glucose metabolism were exclusive to either T1D or HC NETomes in response to PMA or ionomycin. For instance, PGM2, MDH1, and RPIA were present in the NETome of PMA-stimulated HC neutrophils, which are essential for glycolysis, TCA cycle, and the pentose phosphate pathway (PPP), respectively, while IDH1 and MDH2, both essential to the TCA cycle pathway, were identified in the NETome of PMA-stimulated T1D neutrophils. 4-AGL and ATP-dependent 6-PFKL, involved in glycogen metabolism and glycolysis, respectively, were present in the NETome of ionomycin-stimulated HC neutrophils. They regulate the breakdown of glucose and subsequent pathways essential in the generation of ATP, a primary source of energy for the rapidly mobilized effector functions of neutrophils. To meet these high energy requirements, neutrophils also employ other metabolic pathways such as OXPHOS and lipid metabolism [34,35,36]. OXPHOS proteins such as NADK were exclusively present in the NETome of PMA-stimulated T1D neutrophils, and ALDH16A1 exclusively in the NETome of PMA-stimulated HC neutrophils, as well as lipid metabolic proteins APOB, ALOX5, and APOBR [37,38,39]. These data demonstrated qualitative differences in the NETome of T1D neutrophils compared to that of HC neutrophils.

In addition, protein abundance in the NETome might affect neutrophil functions. Upon NET protein quantification, T1D subjects clustered differently compared to HC donors in both PMA- and ionomycin-stimulated conditions. Proteins, such as S100A6, S100P, ELANE and SERPINA3, enriched in the HC NETome in response to either PMA or ionomycin, are involved in neutrophil immunity, confirming the functional integrity of HC neutrophils. On the other hand, the majority of proteins enriched within the T1D NETome were involved in glycolysis, gluconeogenesis, and glucose metabolism. ALDOA, GAPDH, PGAM1, PKM, and PGK1 catalyze important steps in the process of glycolysis and gluconeogenesis, whereas UGP2 is involved in the production of glycogen, suggesting a possible alteration in glucose metabolism of T1D neutrophils.

The ability of neutrophils to rapidly deploy various anti-microbial effector functions, such as NETosis, in response to inflammation and injury, points to both functional plasticity and metabolic heterogeneity. Glucose uptake and glycolysis are essential for neutrophils to NET and to be able to respond properly to various pathogens [32]. We investigated whether the enrichment in metabolic proteins in the T1D NETome reflected alterations in the bioenergetic profile of the T1D neutrophils. Surprisingly the rate of ECAR, a qualitative indicator of glycolysis, was similarly increased in T1D and HC neutrophils in response to PMA or ionomycin. Moreover, lactate levels, at baseline and in response to the stimuli, were also comparable in supernatants of neutrophil cultures of HC and T1D subjects. Of note, ECAR can be influenced by factors other than glycolysis-associated lactate production [40]. Nevertheless, the ‘normal’ bioenergetic profile of T1D neutrophils, along with the enrichment in metabolic proteins in the T1D NETome, suggest that T1D neutrophils have altered the abundance of metabolic proteins, possibly to avoid metabolic and functional impairment. However, in the context of autoimmune diseases, alterations in neutrophil metabolism have been shown to be associated with modified effector functions that could contribute to disease pathology. In the case of active RA, increased neutrophil glycolysis was associated with enhanced NET formation [41]. This increase was shown to occur with the help of hypoxia-inducible transcription factor (HIF-1α), which is stabilized by glycolysis [42] and acts by regulating key glycolytic enzymes in environments poor in oxygen, such as the synovial fluid of RA patients [43]. On the other hand, SLE neutrophils had impaired glycolytic flux and decreased NADPH oxidase-dependent ROS production, which is associated with increased mitochondrial ROS and NETosis [41]. As such, neutrophils with distinct functional and metabolic traits may be characteristic of different types of autoimmune diseases. Taken together, these studies indicate that the implications of an altered metabolic profile in neutrophils may be dependent on disease stage and presentation. Further studies into the metabolic and bioenergetic profile of T1D neutrophils are needed to better understand how neutrophil behavior is impacted during T1D development and how this may contribute to disease pathology.

In summary, our study demonstrated that neutrophils of people with established T1D had a lower abundance of proteins involved in innate immunity, irrespective of normal frequencies of NETing neutrophils and levels of circulating NET constituents. Interestingly, the enrichment of proteins involved in glucose metabolism in the T1D NETome was accompanied by normal bioenergetic profiles measured by extracellular flux analysis (Figure 9). These ‘metabolically stable’ neutrophils with an altered NETome might point to a possible adaptation mechanism to avoid functional impairment. Whilst it is clear that neutrophils are not intrinsically abnormal in T1D individuals, they are involved in different aspects of T1D pathophysiology. The implications of the current findings on neutrophil characteristics and functions, including NETosis, in T1D, warrant further investigation. A better understanding of how neutrophils are affected during the different disease stages of T1D might shed light on the involvement of neutrophils and different types of NETs in T1D development and progression.

## Figures and Tables

**Figure 1 cells-12-01319-f001:**
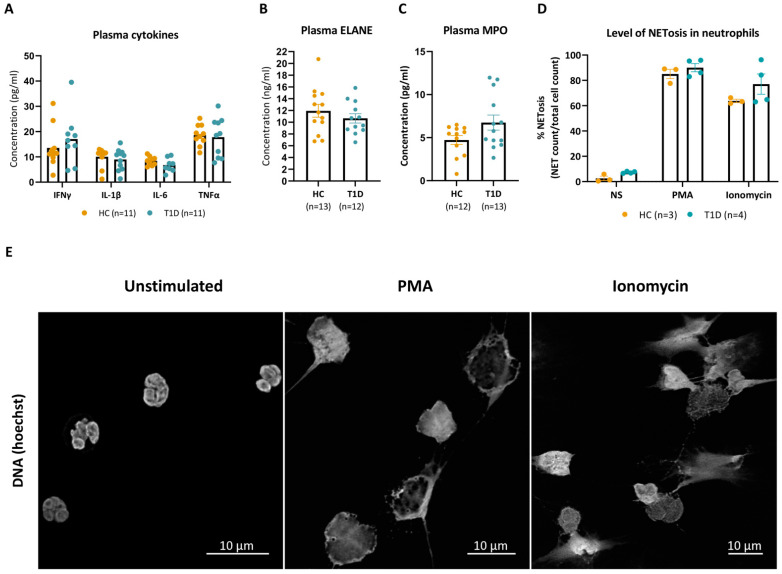
PMA and ionomycin induced NET formation in HC and T1D neutrophils at comparable levels. (**A**) Levels of IFNγ, IL-1β, IL-6 and TNFα (pg/mL) in plasma of HC and T1D subjects. (**B**,**C**) Neutrophil elastase levels (ELANE, ng/mL, (**B**) and myeloperoxidase (MPO, pg/mL, (**C**) levels in the plasma of T1D and HC subjects. Data points below the fit curve of the MSD assays are not represented. (**D**) Percentages (%) of NET formation in unstimulated, PMA- and ionomycin-stimulated conditions in neutrophils of T1D and HC subjects were determined by dividing the counts of NETing cells over total cells. Cells were counted manually up to 500 per slide, with the conditions blinded. (**A**–**C**) HCs are represented with yellow dots, and T1D with blue dots. Error bars represent mean ± SEM, unpaired Mann–Whitney test. (**E**) Representative immunofluorescence images of HC neutrophil DNA stained with Hoechst after a 3 h incubation without stimuli (unstimulated, left panel), with phorbol-myristate acetate (PMA, middle panel) or ionomycin (right panel).

**Figure 2 cells-12-01319-f002:**
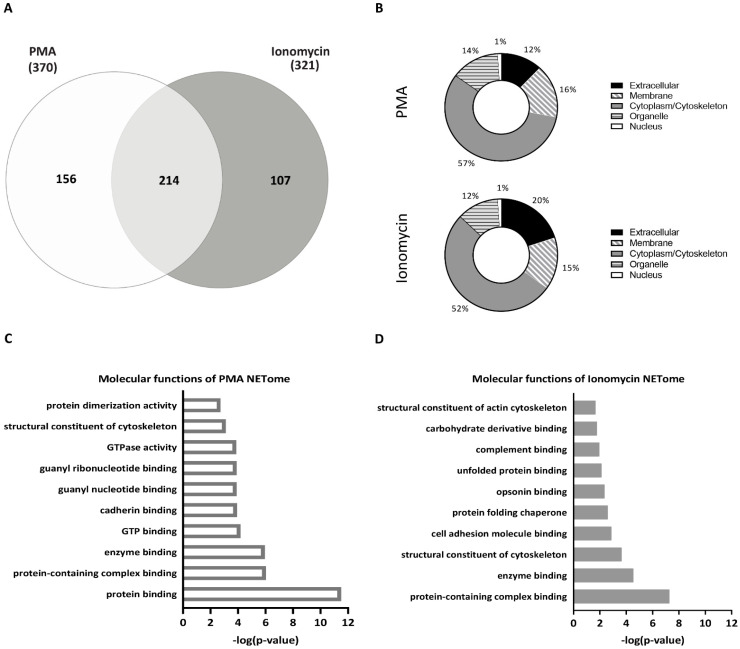
NET proteomes of HC and T1D neutrophils stimulated with PMA and ionomycin. (**A**) Venn diagram of overlapping and unique NET proteins in combined HC and T1D samples (HC, n = 3; T1D, n = 4) after PMA (left, white) and ionomycin (right, grey) stimulation. The number of proteins identified using LC-MS/MS is indicated (<1.0% FDR). (**B**) GO cellular component analysis of NET proteins identified in PMA- (top panel) and ionomycin- (bottom panel) stimulated neutrophils (combined HC and T1D samples). Percentages (%) of proteins belonging to each cellular component are indicated. (**C**,**D**) Graphs representing top 10 most significant pathways associated with NET proteins exclusive to neutrophils stimulated with PMA (**C**) or ionomycin (**D**) (Reactome analysis). The pathways are indicated on the *y*-axis, and their *p*-value (−log10)—on the *x*-axis.

**Figure 3 cells-12-01319-f003:**
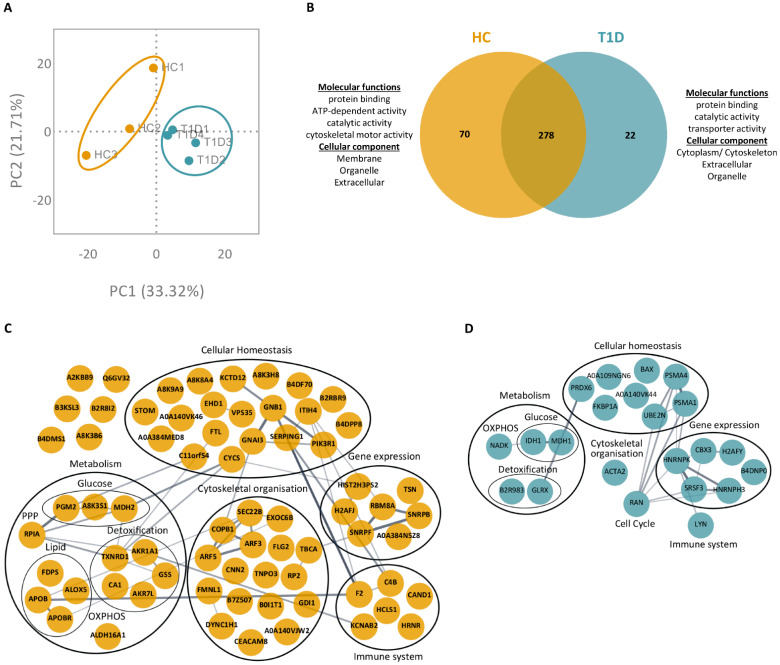
NET proteome of PMA-stimulated T1D neutrophils is distinct from those of HC. (**A**) Principal component analysis (PCA) of all proteins identified in the NETomes of HC (yellow, n = 3) and T1D (blue, n = 4) subjects in response to PMA (<1.0% FDR). (**B**) Venn diagram of overlapping and unique NET proteins in PMA-stimulated HC (left, yellow) and T1D (right, blue) neutrophils. GO annotations of molecular functions and cellular components of proteins exclusive to either HC (left) or T1D (right) NETomes are indicated. (**C**,**D**) Protein–protein interaction networks of proteins exclusive to either HC (**C**) or T1D (**D**) NETomes, in response to PMA. Proteins are represented by their UniProt gene names. Proteins without assigned UniProt gene names are represented by their unique accession numbers. The thickness of an edge is proportional to the confidence score of the protein–protein interaction. When possible, proteins are clustered and annotated according to the primary biological process in which they are involved (GO).

**Figure 4 cells-12-01319-f004:**
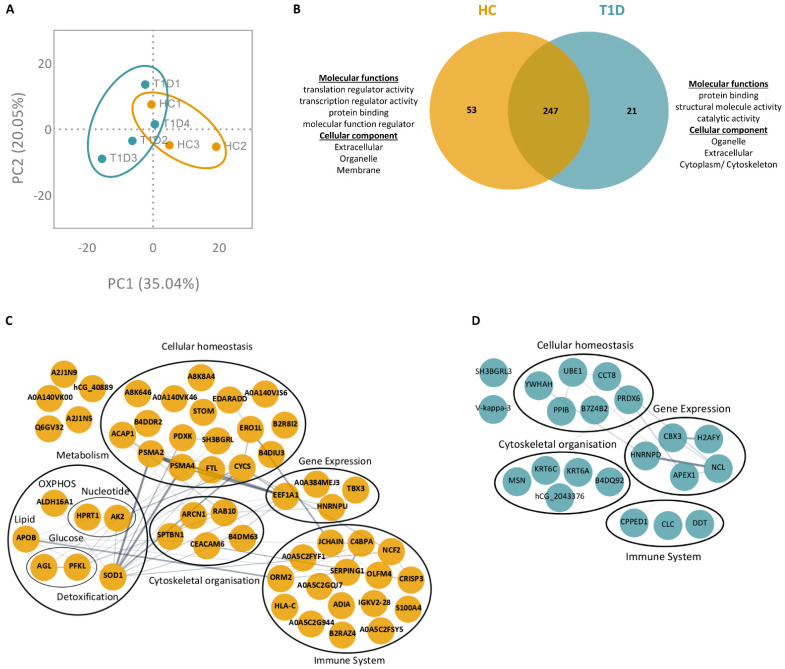
NET proteome of ionomycin-stimulated T1D neutrophils is distinct from those of HC. (**A**) Principal component analysis (PCA) of all proteins identified in the NETomes of HC (yellow, n = 3) and T1D (blue, n = 4) subjects in response to ionomycin (<1.0% FDR). (**B**) Venn diagram of overlapping and unique NET proteins in ionomycin-stimulated HC (left, yellow) and T1D (right, blue) neutrophils. GO annotations for molecular functions and cellular components of proteins exclusive to either HC (left) or T1D (right) NETomes are indicated. (**C**,**D**) Protein–protein interaction networks of proteins exclusive to either HC (**C**) or T1D (**D**) NETomes, in response to ionomycin. Proteins are represented by their UniProt gene names. Proteins without assigned UniProt gene names are represented by their unique accession numbers. The thickness of an edge is proportional to the confidence score of the protein–protein interaction. When possible, proteins are clustered and annotated according to the primary biological process in which they are involved (GO).

**Figure 5 cells-12-01319-f005:**
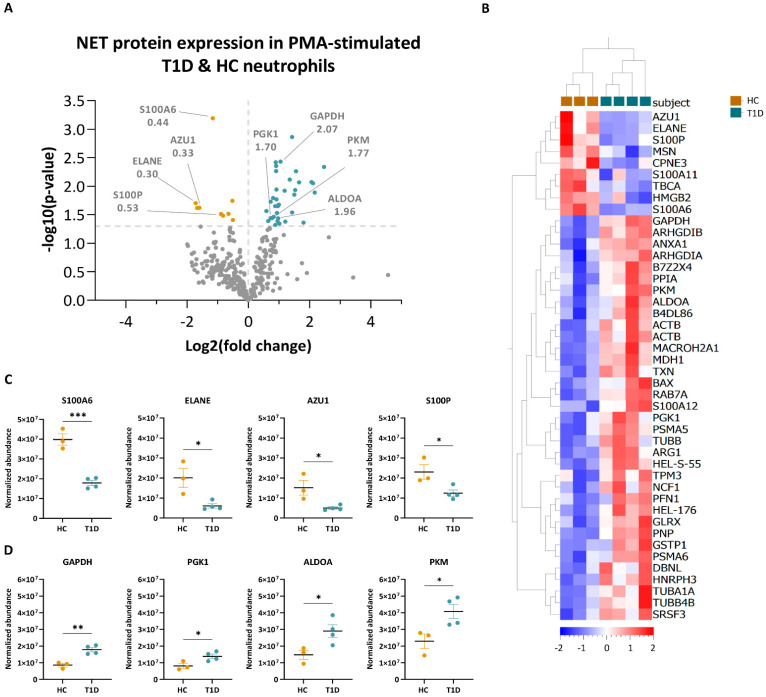
Quantification of NET proteins from PMA-stimulated HC and T1D neutrophils. (**A**) Volcano plot based on fold change (log2) and *p*-value (−log10) of all differentially expressed NET proteins identified in NETing neutrophils of HC (n = 3) and T1D (n = 4) subjects in response to PMA (<1.0% FDR). Th threshold (grey dotted line) was set at *p*-value < 0.05. Proteins below the threshold are in grey, while proteins with higher expression levels in T1D subjects are in blue, and those in HC are in yellow. Proteins of interest, as well as their fold change values, are indicated. (**B**) Hierarchical clustering of normalized abundance values (Progenesis) of significantly different NET proteins of HC (yellow) and T1D (blue) subjects after PMA stimulation (<1.0% FDR, *p* < 0.05, unpaired two-tailed *t*-test). Clustering was performed on subjects (columns) and proteins (rows). Scale bar represents the normalized abundance values. (**C**,**D**) Normalized abundance values of NET proteins enriched in HC (**C**) and T1D (**D**) neutrophils in response to PMA. Abbreviations: S100A6: protein S100-A6; ELANE: neutrophil elastase; AZU1: azurocidin; S100P: protein S100-P; GAPDH: glyceraldehyde-3-phosphate dehydrogenase; PGK1: phosphoglycerate kinase; ALDOA: fructose-bisphosphate aldolase A; PKM: pyruvate kinase. Error bars represent mean SEM (n = 3 for HC and n = 4 for T1D). Unpaired two-tailed *t*-test. * *p*-value < 0.05, ** *p*-value < 0.01, *** *p*-value < 0.001.

**Figure 6 cells-12-01319-f006:**
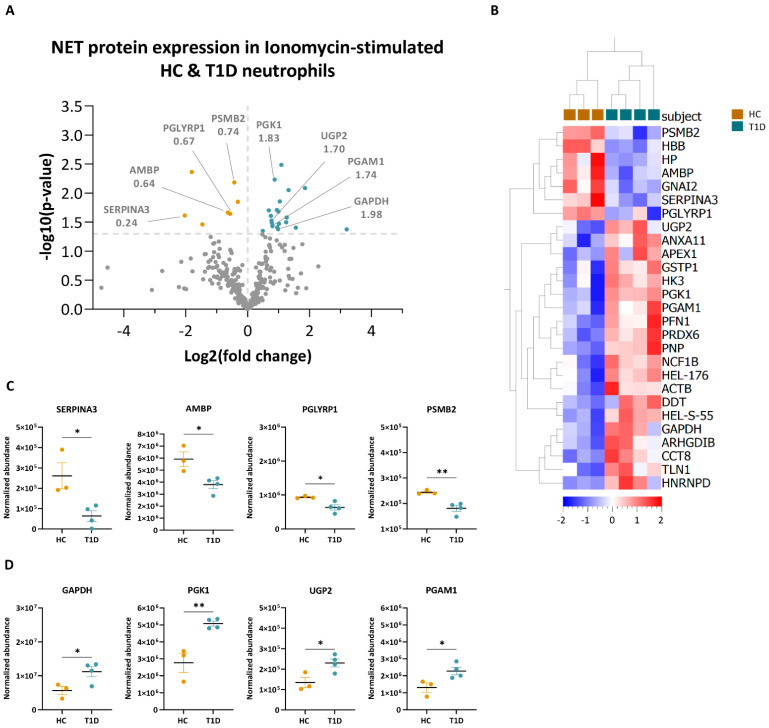
Quantification of NET proteins from ionomycin-stimulated HC and T1D neutrophils. (**A**) Volcano plot based on fold change (log2) and *p*-value (−log10) of all differentially expressed NET proteins identified in NETing neutrophils of HC (n = 4) and T1D (n = 3) subjects in response to ionomycin (<1.0% FDR). Threshold (grey dotted line) was set at *p*-value < 0.05. Proteins below the threshold are in grey, while proteins with higher expression levels in T1D subjects are in blue, and those in HC are in yellow. Proteins of interest, as well as their fold change values, are indicated. (**B**) Hierarchical clustering of normalized abundance values (Progenesis) of significantly different NET proteins of HC (yellow) and T1D (blue) subjects after ionomycin stimulation (<1.0% FDR, *p* < 0.05, unpaired two-tailed *t*-test). Clustering was performed on subjects (columns) and proteins (rows). Scale bar represents the normalized abundance values. (**C**,**D**) Normalized abundance values of NET proteins enriched in HC (**C**) and T1D (**D**) neutrophils in response to ionomycin. Abbreviations: SERPINA3: alpha-1-antichymotrypsin; AMBP: protein AMBP; PSMB2: proteasome subunit beta; PGLYRP1: peptidoglycan recognition protein 1; GAPDH: glyceraldehyde-3-phosphate dehydrogenase; PGK1: phosphoglycerate kinase; UGP2: UTP-glucose-1-phosphate uridylyltransferase; PGAM1: phosphoglycerate mutase 1. Error bars represent mean SEM (n = 3 for HC and n = 4 for T1D). Unpaired two-tailed *t*-test. * *p*-value < 0.05, ** *p*-value < 0.01.

**Figure 7 cells-12-01319-f007:**
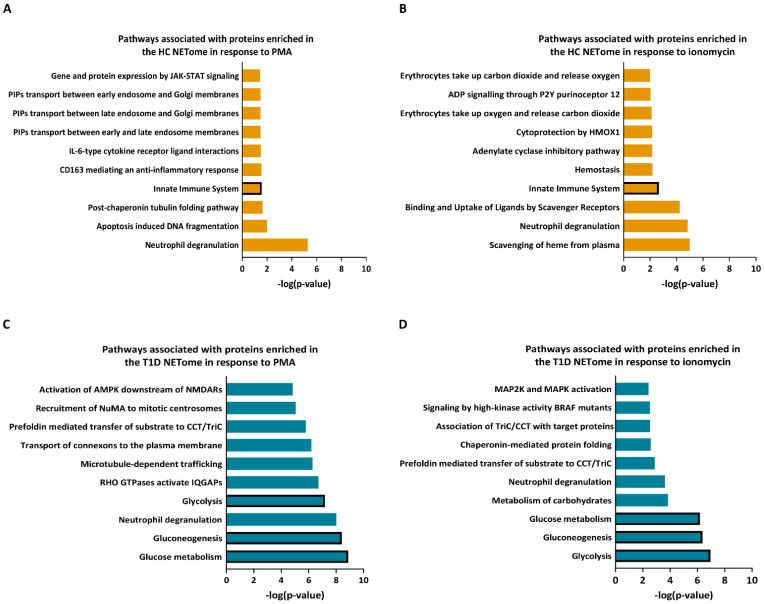
Pathway analysis of NET proteins enriched in HC and T1D neutrophils in response to PMA and ionomycin. (**A**,**B**) Graphs representing top 10 most significant pathways associated with NET proteins with higher expression in neutrophils of HC subjects following PMA (**A**) and ionomycin (**B**) stimulation. (**C**,**D**) Graphs representing top 10 most significant pathways associated with NET proteins with higher expression in neutrophils of T1D subjects following PMA (**C**) and ionomycin (**D**) stimulation (Reactome analysis). The pathways are indicated on the *y*-axis, and their *p*-value (−log10)—on the *x*-axis. Bars representing the relevant pathways are outlined in bold.

**Figure 8 cells-12-01319-f008:**
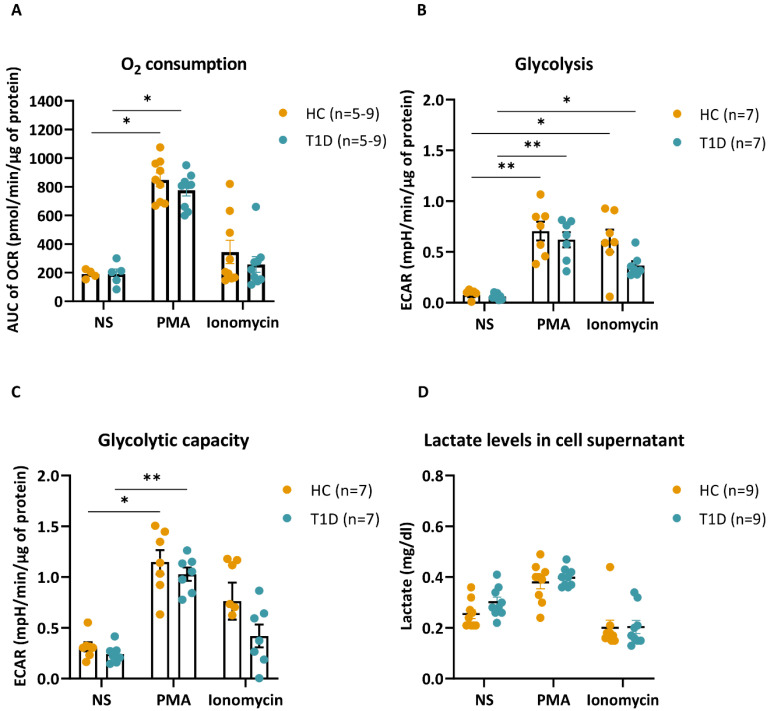
Bioenergetic profile of HC and T1D neutrophils in response to PMA and ionomycin. (**A**) Graph representing the area under the curve (AUC) of normalized oxygen consumption rate (OCR, pmol/min/µg of protein) in unstimulated (NS), PMA- or ionomycin-stimulated neutrophils of HC (n = 5–9) and T1D (n = 5–9) subjects. Bar graphs show mean ± SEM. (**B**,**C**) Extracellular acidification rate (ECAR)-based calculation of glycolysis (**B**) and glycolytic capacity (**C**) in unstimulated (NS), PMA- or ionomycin-stimulated HC (n = 7) and T1D (n = 7) neutrophils. Bar graphs show mean ± SEM. (**D**) Lactate levels (mg/dL) in the supernatants of unstimulated (NS) and PMA- or ionomycin-stimulated HC (n = 9) and T1D (n = 9) neutrophils. Error bars show SEM (n = 9). Kruskal–Wallis test with Dunn’s post hoc test. * *p*-value < 0.01, ** *p*-value < 0.001.

**Figure 9 cells-12-01319-f009:**
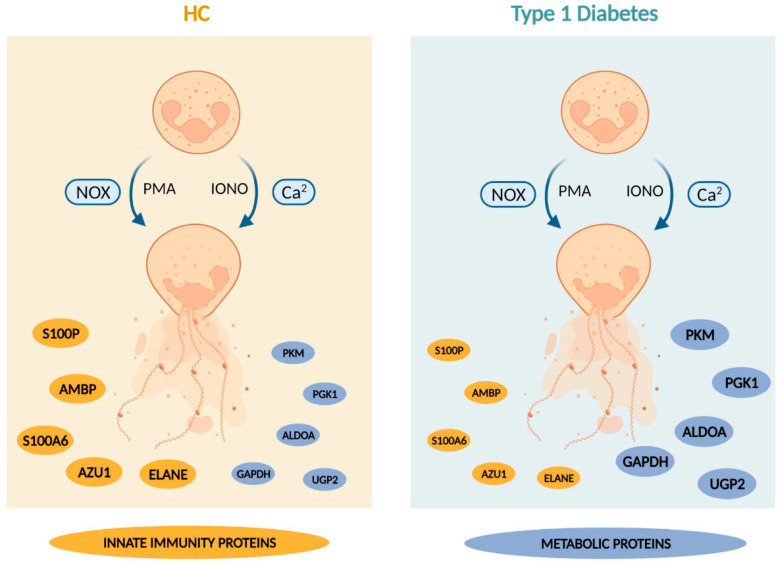
Summary Scheme summarizing our main findings (created with Biorender.com). Briefly, our study demonstrated that neutrophils of people with established T1D undergo NET formation at rates similar to those of HC subjects in response to PMA and ionomycin. Furthermore, we showed that the T1D NETomes, induced by these stimuli, had an enrichment in proteins involved in glucose metabolism, with lower abundances in proteins implicated in innate immunity, compared to those of HC. This altered NETome composition was not associated with higher rates of glycolysis or glycolytic capacity compared to HC. The size of the nodules representing the proteins reflects the protein abundance in HC and T1D NETomes. Abbreviations: HC: healthy control; T1D: type 1 diabetes; NOX: nicotinamide adenine dinucleotide phosphate (NADPH) oxidase; Ca^2^: calcium; PMA: phorbol-myristate acetate; IONO: ionomycin; S100P: protein S100-P; AMBP: protein AMBP; S100A6: protein S100-A6; AZU1: azurocidin; ELANE: neutrophil elastase; PKM: pyruvate kinase; PGK1: phosphoglycerate kinase; ALDOA: fructose-bisphosphate aldolase A; GAPDH: glyceraldehyde-3-phosphate dehydrogenase; UGP2: UTP-glucose-1-phosphate uridylyltransferase.

**Table 1 cells-12-01319-t001:** Patient and healthy control demographics.

	Type 1 Diabetes	Healthy Control
Number of donors	14	12
Age (years) *	36.3 (19–61)	34.5 (24–65)
Disease duration (years) *	17.8 (4–38)	na
Gender F/M	8/6	5/7
Glycemia (mg/dL) *	132.7 (76–175)	na
HbA1c (%) *	7.8 (5.9–13.7)	na
Time In Range (TIR, %) *	53.5 (19–88)	na
Insulin total daily dose (U/day) *	33.8 (24.9–84)	na
GADA (U/mL) pos/neg **	5/9	0/12
IA2-A (U/mL) pos/neg ***	5/9	0/12
IAA (% binding) pos/neg ****	11/3	0/12
ZnT8A (% binding) pos/neg *****	1/13	0/12

na (not applicable); * mean (range); ** glutamic acid decarboxylase autoantibody titers (<23); *** insulinoma-associated protein 2 autoantibody titers (<1.4); **** insulin autoantibody titers (<0.6); ***** zinc transporter 8 autoantibody titers (<1.01).

**Table 2 cells-12-01319-t002:** Non-exhaustive list of prototypic neutrophil extracellular trap (NET) proteins identified by proteomic analysis in unstimulated, PMA- and ionomycin-stimulated conditions in combined T1D and HC samples.

Cellular Localization	Protein Name	Gene Name	Accession Number
Granules	Azurocidin	*AZU1*	P20160
	Cathepsin G	*CTSG*	P08311
	Lactotransferrin	*LTF*	P02788
	Myeloperoxidase	*MPO*	P05164
	Neutrophil defensin 1	*DEFA1*	P59665
	Neutrophil Elastase	*ELANE*	P08246
Nucleus	Neutrophil gelatinase-associated lipocalin	NGAL	B2ZDQ1
	Histone H2B	*HIST1H2BK*	O60814
	Myeloid cell nuclear differentiation antigen	*MNDA*	P41218
	Protein-arginine deiminase type-4	*PADI4*	Q9UM07
Cytoplasm	Protein S100-A6	*S100A6*	P06703
	Protein S100-A8	*S100A8*	P05109
	Protein S100-A9	*S100A9*	P06702
	Protein S100-A11	*S100A11*	P31949
	Protein S100-A12	*S100A12*	P80511
	Protein S100-P	*S100P*	P25815
Cytoskeletal	Actin (cytoplasmic)	*ACTB*	P60709
	Myosin-9	*MYH9*	P35579
Peroxisomal	Catalase	*CAT*	P04040

## Data Availability

The mass spectrometry proteomics data have been deposited to the ProteomeXchange Consortium via the PRIDE [44] partner repository with the dataset identifier PXD033599 and 10.6019/PXD033599.

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
