# Peer review of "NET Proteome in Established Type 1 Diabetes Is Enriched in Metabolic Proteins"

_cells, 2023, doi:10.3390/cells12091319_

Round 1

Reviewer 1 Report (Previous Reviewer 3)

The authors provided the responses to my comments, and in my opinion the manuscript is suitable for publication 

Author Response

We thank the reviewer for his/her positive evaluation of the manuscript. 

Reviewer 2 Report (New Reviewer)

NETOsis and neutrophils have been suggested to play a role in T1DM development, however, with contradictory findings. Bissenova et al undertook an extensive proteomic analysis of NETome in neutrophils isolated from patients with etsbalished T1DM compared to nondiabetic individuals. Both neutrophils from healthy controls and from T1DM patients were characterized by a similar magnitude of response to NETosis activators as well as by similar bioenergetic profiles. Interestingly, the NETome of T1DM patients was siginifincatly enriched in proteins involved in glycolysis and glucose metabolis with a parallel decrease in proteins involved in innate immunity. The results indicate that the neutrophil proteome and, particularly, NETome undergo alterations during T1DM development that could influence the disease pathophysiology. 

I have minor comments:

1. Please indicate in the text (l 399-401) and in Fig.1E legend/description that only NETosis in neutrophils of HC is shown:

"Staining with the DNA-intercalating dye Hoechst identified mesh-like DNA structures, indicative of the formation of NETs, in PMA- or ionomycin-stimulated neutrophils (figure 1E)".

2. It is unclear what Fig.2 shows. Are these changes observed in HC or in T1DM neutrophil samples? Or is it pooled data from all (HC+T1DM) samples? Please clearly describe in the text  and in Fig.2 legend.

A similar clarification is needed in the title of Table 2.

Author Response

Response 1: We thank the reviewer for bringing our attention to this. We have now revised the aforementioned sentence (p. 8, lines 266-268)

In the figure 1E legend, we specify: “(E) Representative immunofluorescence images of HC neutrophil DNA stained with Hoechst after a 3-hour incubation without stimuli (unstimulated, left panel), with phorbol-myristate acetate (PMA, middle panel) or ionomycin (right panel).”

Response 2: Indeed, in Figure 2 we combined the results from HC and T1D samples in order to firstly investigate the NET proteome differences between the two stimuli (PMA and ionomycin). In Table 2, we summarized the NET proteins found in all of the samples, irrespective of stimuli or donor.

We have now clarified this in the text (p.8, lines 275-276) and in the figure/table legends (p.7, lines 243-251; p. 8, lines 285-286).

This manuscript is a resubmission of an earlier submission. The following is a list of the peer review reports and author responses from that submission.

Round 1

Reviewer 1 Report

The topic of the publication is very topical. Currently, a lot of research is conducted aimed at finding methods to prevent the development of glucose homeostasis disorders.

Author Response

Point 1: The topic of the publication is very topical. Currently, a lot of research is conducted aimed at finding methods to prevent the development of glucose homeostasis disorders.

Response 1: We thank the reviewer for his/her positive evaluation of the manuscript. The role of neutrophils in the field of diabetes is just recently being explored.

Reviewer 2 Report

This is a nice manuscript regarding the NET proteome in persons type 1 diabetes. This article is interesting but exceedingly long. First, I suggest shortening it to attract more readers and to be more easily readable.

Additionally, table captions should have all the relevant information including abbreviation explanation, for the tabel to be understandable as a standalone unit. There are too many abbreviations and a redar might get confused. Please omit al unnecessary ones. Particularly, avoid abbreviations in titles and subtitles.

Abstract

Please provide the structured abstract. The results must be more clearly emphasized in the abstract and their meaning for clinical practice mentioned.

Introduction

Introduction is too long. In the "aim of the study" part, it already reflects the results of the study. Please omit that and bear in mind, that the whole introduction must be concise and composed of only three paragraphs. 

Methods

From this section, it si not clear, how many patients were included. Please provide more straight forward description. Also, explain the protocol of the study in detail. When was the blood collected etc.

This part must be shortened significantly. 

Results

Please avoid comments on previous research in the results section. Please provide only your own results, as comments are for the discussion section. Also, do not explain the procedures of the experiment in the results section. Describe results only.

Discussion

Discussion is too long, too descriptive, and unfocused. Please shorten the discussion and focus on finding and their relevance also focus on potential clinical relevance. This is not clear from the current discussion. Also, please provide the results of other findings and omit descriptive part where you describe what you did several times.

Author Response

Point 1: This is a nice manuscript regarding the NET proteome in persons type 1 diabetes. This article is interesting but exceedingly long. First, I suggest shortening it to attract more readers and to be more easily readable.

Response 1: We thank the reviewer for his/her constructive comments. We hope that by shortening the manuscript and keeping the most valuable data we have made the overall message stronger for a broader audience.

Point 2: Additionally, table captions should have all the relevant information including abbreviation explanation, for the table to be understandable as a standalone unit. There are too many abbreviations and a reader might get confused. Please omit all unnecessary ones. Particularly, avoid abbreviations in titles and subtitles.

Response 2: We edited all tables and their respective captions according to the reviewer’s suggestions (Tables 1-3 on p. 4 and 10). We omitted unnecessary abbreviations such as ‘LC-MS/MS’ and ‘NET’ and added explanations to the necessary abbreviations.

Point 3: Abstract. Please provide the structured abstract. The results must be more clearly emphasized in the abstract and their meaning for clinical practice mentioned.

Response 3: We now provide a structured and concise abstract that focuses on the key findings of the paper and their clinical implications (p. 1 lines 19-42).

Point 4: Introduction. Introduction is too long. In the "aim of the study" part, it already reflects the results of the study. Please omit that and bear in mind, that the whole introduction must be concise and composed of only three paragraphs. 

Response 4: We thank the reviewer for this helpful suggestion. The introduction was considerably shortened and we omitted elaborations on the results of the study (p. 2).

Point 5: Methods. From this section, it is not clear, how many patients were included. Please provide more straight forward description. Also, explain the protocol of the study in detail. When was the blood collected etc. This part must be shortened significantly. 

Response 5: We refer the reviewer to tables 1-2 (p.4) in which we clarified the number of healthy controls and patients listed under ‘Number of individuals’. Blood was always drawn in the morning (between 8:00 and 10:00 am) taking into account circadian variations of neutrophil biology and all experiments are performed with freshly isolated neutrophils. This has been clarified on p 4. line 129. We also significantly shortened certain paragraphs in the Methods section according to the reviewer’s suggestion.

Point 6: Results. Please avoid comments on previous research in the results section. Please provide only your own results, as comments are for the discussion section. Also, do not explain the procedures of the experiment in the results section. Describe results only.

Response 6: We adapted the Results section by removing all references to previous research, as well as descriptions of experimental procedures. Notably those on p. 8 line 284-285, p. 9-10 lines 317-322, p.10 lines 338-342 and p. 19 lines 525-528. Additionally, we edited Figure 2 by moving the scheme describing the NET protein preparation for the proteomics analysis to the Supplementary Figures section (p. 9).

Point 7: Discussion. Discussion is too long, too descriptive, and unfocused. Please shorten the discussion and focus on findings and their relevance also focus on potential clinical relevance. This is not clear from the current discussion. Also, please provide the results of other findings and omit descriptive part where you describe what you did several times.

Response 7:  We revised the manuscript to the suggestions of the reviewer. We omitted lengthy descriptions of our findings and focused specifically on their implications and clinical relevance (p. 19-21).

Reviewer 3 Report

Dear authors,

the manuscript is very interesting because focused on the aspects of neutrophils biology that affects the pathophysiology of T1. 

I have some concerns:

1. The results are showed in good manner, but the authors have to indicate in each figure the N=? related to each experiments.

2. What is the criterion for the choice of patients destined to spectrometry or to bioenergetics experiments? Hb1Ac is very different for groups (LC vs bioenergetic) and this could introduce a bias for a good interpretation of the global results. The authors should be consider the "time in range" of patients.

3. The lacking of an ex-vivo validation - in a much larger cohort of T1D - of the main proteins found in omic experiments, e.g. metabolic proteins.

4. the ELANE levels in the group of T1D neutrophils vs HC (figure 1) would include the statistical significance between groups). It means is not significant?

Author Response

Point 1: The results are showed in good manner, but the authors have to indicate in each figure the N=? related to each experiments.

Response 1: We thank the reviewer for the insightful remark. The revised manuscript now contains, in the each figure and figure legend, the number of healthy control and patient donors used for the experiment. 

Point 2: What is the criterion for the choice of patients destined to spectrometry or to bioenergetics experiments? Hb1Ac is very different for groups (LC vs bioenergetic) and this could introduce a bias for a good interpretation of the global results. The authors should be consider the "time in range" of patients.

Response 2: For both experiments we have chosen people with established T1D (>4 years after disease onset). As for the HbA1c levels, the T1D group of the proteomic analysis had the following values: 8.2, 8.2, 7 and 13.7%, whereas those of the bioenergetic analysis experiment had the following values: 7.4, 6.3, 8.8, 5.9, 7.9, 6.4, 6.1, 6.1, 6.8 and 6.8%. Therefore, the main variability between the two groups is due to the T1D individual with an HbA1c value of 13.7%. However, both PCAs (Figures 3A, 4A) and the hierarchical clustering in the heatmaps (Figures 5B, 6B) demonstrated that the patient samples in the proteomic analysis clustered together. This points to the relative homogeny of the samples in the T1D group despite the differences in HbA1c. Additionally, we have included, as suggested by the reviewer, time in range (TIR) values in Table 1 and 2 (p. 4).

Point 3: The lacking of an ex-vivo validation - in a much larger cohort of T1D - of the main proteins found in omic experiments, e.g. metabolic proteins.

Response 3: We agree with the reviewer that a larger validation cohort would strengthen the overall results of the study. However, in the scope of this manuscript, we focused on the pathways associated with proteins enriched in the T1D NETome in response to both PMA and ionomycin. Different analyses pointed towards metabolic pathways (i.e. glycolysis, glyconeogenesis, and glucose metabolism).

Point 4: The ELANE levels in the group of T1D neutrophils vs HC (figure 1) would include the statistical significance between groups). It means is not significant?

Response 4: The reviewer is correct, the mean levels of ELANE in HC and T1D groups were not statically significant (unpaired Mann-Whitney test, figure legend lines 301-302). We refer the reviewer to the Results section p. 8 lines 287-290.

Round 2

Reviewer 3 Report

I think that this study shows only preliminary results.

First of all, the healthy controls are true controls? what is the method for attesting the classification as control? what their glycemia? please, report on the table 1 and 2 this information.

Fig. 1 In this work the validation on serum or plasma in a large cohort of T1D or controls are lacking. The measurements of cytokines, elane, and other cannot be done only on 4 T1D. As I reported previously, the sample size is very important and for the assertion of the prevalence (or not) of metabolic proteins need to measure in the samples.  

Above the bioinformatics, the authors did not assume to validate their results in a large cohort of T1D with adeguate sample size. Before to submit to their ethic board, how do you calculate sample size?